# The MLL3/4 complexes and MiDAC co-regulate H4K20ac to control a specific gene expression program

Xiaokang Wang[1,3,*], Wojciech Rosikiewicz[2,*], Yurii Sedkov[1,*], Baisakhi Mondal[1], Tanner Martinez[1], Satish Kallappagoudar[1], Andrey Tvardovskiy[3], Richa Bajpai[1], Beisi Xu[2], Shondra M Pruett-Miller[1], Robert Schneider[3], Hans-Martin Herz[1]

**The mitotic deacetylase complex MiDAC has recently been shown to play a vital physiological role in embryonic development and neurite outgrowth. However, how MiDAC functionally intersects with other chromatin-modifying regulators is poorly understood. Here, we describe a physical interaction between the histone H3K27 demethylase UTX, a complex-specific subunit of the enhancer-associated MLL3/4 complexes, and MiDAC. We demonstrate that UTX bridges the association of the MLL3/4 complexes and MiDAC by interacting with ELMSAN1, a scaffolding subunit of MiDAC. Our data suggest that MiDAC constitutes a negative genome-wide regulator of H4K20ac, an activity which is counteracted by the MLL3/4 complexes. MiDAC and the MLL3/4 complexes co-localize at many genomic regions, which are enriched for H4K20ac and the enhancer marks H3K4me1, H3K4me2, and H3K27ac. We find that MiDAC antagonizes the recruitment of UTX and MLL4 and negatively regulates H4K20ac, and to a lesser extent H3K4me2 and H3K27ac, resulting in transcriptional attenuation of associated genes. In summary, our findings provide a paradigm how the opposing roles of chromatin-modifying components, such as MiDAC and the MLL3/4 complexes, balance the transcriptional output of specific gene expression programs.**

## Introduction

MLL3 (also known as KMT2C) and MLL4 (also known as KMT2D) are chromatin-modifying proteins that monomethylate histone H3K4 via their catalytic SET domains. MLL3 and MLL4 exist in two separate macromolecular complexes that belong to a total of six mammalian complexes of the compositionally and functionally highly conserved COMPASS family (Shilatifard, 2012; Herz, 2016; Cenik & Shilatifard, 2021). All complexes share identical core subunits but

also contain complex-specific subunits that are conserved only within one of three metazoan branches. Each metazoan branch is represented by two mammalian complexes, namely, the SET1A/B complexes (branch one), the MLL1/2 complexes (branch two), and the MLL3/4 complexes (branch three). UTX (also known as KDM6A) exists as a complex-specific subunit within the MLL3/4 complexes and acts as a histone H3K27 demethylase removing methyl groups from the inhibitory histone marks H3K27me3 and H3K27me2 via its Jumonji C domain (Hong et al, 2007; Lee et al, 2013). Our previous studies and the work of others have demonstrated that the MLL3/4 complexes function as major H3K4 monomethyltransferases on enhancers, providing a model in which prior removal of H3K27me3/2 via UTX is required on inactive or poised enhancers, before they can transition to an activated state via addition of H3K4me1 through MLL3/4 (Herz et al, 2010, 2012; Hu et al, 2013; Lee et al, 2013; Rickels et al, 2017). Despite our increasing understanding of UTX, MLL3, and MLL4 in regulating enhancer activity in development and disease, many functional aspects of the MLL3/4 complexes remain ill-defined.

Lysine residues on histones can also be acetylated by histone acetyltransferases (HATs) and deacetylated by histone deacetylases (HDACs) (Shahbazian & Grunstein, 2007). Histone acetylation has been found on all four core histones (H1-4) and can be deposited by specific writers and recognized by site-specific readers at multiple lysine residues on each core histone (Graff & Tsai, 2013; Marmorstein & Zhou, 2014; Barnes et al, 2019). The mammalian genome encodes multiple classes of histone deacetylases, among which the class I histone deacetylases (HDAC1-3, 8) are the most well studied (Milazzo et al, 2020). HDAC1-3 are assembled into large multi-subunit protein complexes to regulate the acetylation state of histones and other non-histone proteins. The integration of HDAC1-3 into these scaffolds both strongly enhances the enzymatic activities and also determines the specificity of these HDAC complexes (Bantscheff et al, 2011; Watson et al, 2012; Millard et al, 2013,

[1]Department of Cell and Molecular Biology, St. Jude Children's Research Hospital, Memphis, TN, USA    [2]Center for Applied Bioinformatics, St. Jude Children's Research Hospital, Memphis, TN, USA    [3]Institute of Functional Epigenetics (IFE), Helmholtz Zentrum München, Neuherberg, Germany

Correspondence: wangx12@chop.edu; hans-martin.herz@stjude.edu
Baisakhi Mondal's present address is Wellcome Trust Centre for Cell Biology, University of Edinburgh, Edinburgh, UK
*Xiaokang Wang, Wojciech Rosikiewicz, and Yurii Sedkov contributed equally to this work.
Xiaokang Wang's present address is Division of Hematology, The Children's Hospital of Philadelphia, Philadelphia, PA, USA

2016; Banks et al, 2020; Turnbull et al, 2020; Wang et al, 2020). HDAC1/2 are integrated into the NuRD, SIN3, CoREST, and MiDAC complexes, whereas HDAC3 is a component of the SMRT/NCoR complex (Laherty et al, 1997; Xue et al, 1998; Li et al, 2000; Oberoi et al, 2011) (reviewed in Millard et al [2017]).

The mitotic deacetylase complex MiDAC, which comprises the scaffolding subunits DNTTIP1, ELMSAN1 (also known as MIDEAS), and the histone deacetylases HDAC1 or HDAC2, was initially identified in a chemoproteomic screen as being specifically enriched on a HDAC-inhibitor-bound resin in cells stalled in G2/prophase of mitosis after nocodazole treatment (Bantscheff et al, 2011). TRERF1 and ZNF541 are paralogs of ELMSAN1 with a more tissue-specific expression pattern and have also been described as scaffolding subunits of MiDAC (Choi et al, 2008; Bantscheff et al, 2011; Hao et al, 2011). MiDAC exhibits a tetrameric architecture with each monomer consisting of DNTTIP1, ELMSAN1, and HDAC1 or HDAC2 thus resembling a three-dimensional X-shape with the HDAC1/2 catalytic sites at the four extremities of the complex suggesting that MiDAC may simultaneously target multiple nucleosomes and may be highly processive (Itoh et al, 2015; Turnbull et al, 2020). MiDAC components are located predominantly in the soluble nuclear fraction throughout the cell cycle and loss of MiDAC function causes misalignment of chromosomes in metaphase (Turnbull et al, 2020). Furthermore, mutations of DNTTIP1 and ELMSAN1 have been identified in different cancer types (Cheng et al, 2017; Piraino & Furney, 2017; Sawai et al, 2018; Xu et al, 2018; Zhang et al, 2018). MiDAC also constitutes an important regulator of a neural gene expression program to ensure proper neuronal maturation and/or neurite outgrowth during neurogenesis (Mondal et al, 2020). Homozygous knock-out mouse embryos lacking either DNTTIP1 or ELMSAN1 die at day E16.5 with severe anemia and a clear malformation of the heart (Turnbull et al, 2020).

Despite recent advances in our understanding of MiDAC's structure and physiological role, it is unknown how MiDAC integrates into chromatin regulatory complexes and pathways. Applying a large-scale interactome analysis, we describe here an association of UTX with MiDAC. We show that UTX and ELMSAN1 form the interface between MiDAC and the MLL3/4 complexes. We demonstrate that the loss of MiDAC function results in a genome-wide increase of H4K20ac, suggesting that H4K20ac constitutes a MiDAC substrate in vivo. The genome-wide increase of H4K20ac in *Dnttip1* KO mouse embryonic stem cells (mESCs) coincides with increased UTX and MLL4 occupancy on many genomic elements, indicating that MiDAC negatively affects the recruitment of the MLL3/4 complexes to chromatin. Whereas H3K4me1 enrichment is slightly decreased, H3K4me2 enrichment is higher at regions displaying increased UTX and MLL4 occupancy, suggesting an activity switch of MLL3/4 towards H3K4me2 in the presence of H4K20ac or absence of MiDAC function. However, in *Mll3/4* double knockout (DKO) mESCs we observe an increase of H3K27me3 at regions with decreased H4K20ac and lower DNTTIP1 occupancy, indicating that H3K27me3 may be involved in suppressing H4K20ac, despite reduced MiDAC recruitment. Taken together, our study reveals for the first time a functional intersection between MiDAC and the MLL3/4 complexes in regulating H4K20ac and describes the antagonistic relationship of these

chromatin-modifying complexes in their role to properly balance transcription of a specific gene expression program.

# Results

## The mitotic deacetylase complex MiDAC associates with the MLL3/4 complexes

To identify novel candidates that regulate the function of the MLL3/4 complexes, we immunoprecipitated UTX, a H3K27 demethylase and complex-specific subunit of the MLL3/4 complexes, from nuclear extracts of human embryonic kidney cells (HEK293 cells). Mass spectrometry (MS) analysis detected as expected all components of the MLL3/4 complexes including all core subunits, complex-specific subunits, and the H3K4 methyltransferases MLL3 and MLL4 (Fig 1A and Table S1). In addition, we also identified two novel UTX interactors: DNTTIP1 and ELMSAN1 (Fig 1A and Table S1). The association of UTX with DNTTIP1 and ELMSAN1 along with other subunits of the MLL3/4 complexes was confirmed by Western blotting (WB) (Fig 1B). Reciprocal immunoprecipitations (IPs) of ELMSAN1 and DNTTIP1 from nuclear extracts of HEK293 cells confirmed an interaction of these proteins with components of the MLL3/4 complexes including UTX, MLL3/4, and the core subunits RBBP5 and ASH2L (Figs 1C and S1). Consistent with previous studies showing that the histone deacetylases HDAC1 and HDAC2 interact with DNTTIP1 and ELMSAN1 to be integrated into the histone deacetylase complex MiDAC (Bantscheff et al, 2011; Mondal et al, 2020; Turnbull et al, 2020), both our ELMSAN1 and DNTTIP1 IPs also purified HDAC1 and HDAC2 (Figs 1C and S1). An association of HDAC1 and HDAC2 with UTX was also observed following UTX IP providing further evidence that UTX associates indeed with MiDAC (Fig 1B and Table S1). Glycerol gradient fractionation of FLAG-affinity purified UTX from nuclear extracts of HEK293 cells confirmed that MiDAC co-migrates with the MLL3/4 complexes (Fig 1D, red box). A substantial amount of MiDAC also associated with UTX outside the MLL3/4 complexes indicating that not all overexpressed UTX is incorporated into the MLL3/4 complexes and that MiDAC might also directly interact with UTX (Fig 1D, blue box). The high number of ELMSAN1 peptides in our UTX IP, as assessed by MS analysis and the fact that after UTX IP a substantial amount of MiDAC co-migrated with UTX outside the MLL3/4 complexes, suggests that UTX might constitute a bridging component between MiDAC and the MLL3/4 complexes (Fig 1D and Table S1). Furthermore, UTX and DNTTIP1 IPs performed against the endogenous proteins in WT and *Mll3/4* DKO mESCs showed that the UTX antibody used could immunoprecipitate UTX and the MiDAC subunits DNTTIP1, ELMSAN1, and HDAC1, whereas our DNTTIP1 IP identified DNTTIP1, ELMSAN1, and HDAC1 as members of MiDAC along with UTX (Fig S2). Thus, these results show the physical association between UTX and ELMSAN1/DNTTIP1 under endogenous conditions and make it unlikely that overexpressed UTX aberrantly associates with MiDAC. Both ELMSAN1 and its paralog TRERF1, but not the paralog ZNF541, are expressed in HEK293 cells (data not shown). Thus, because of a potential redundancy between ELMSAN1 and TRERF1 we generated an *ELMSAN1 TRERF1* double knockout (DKO) HEK293 cell line to establish the interaction interface between

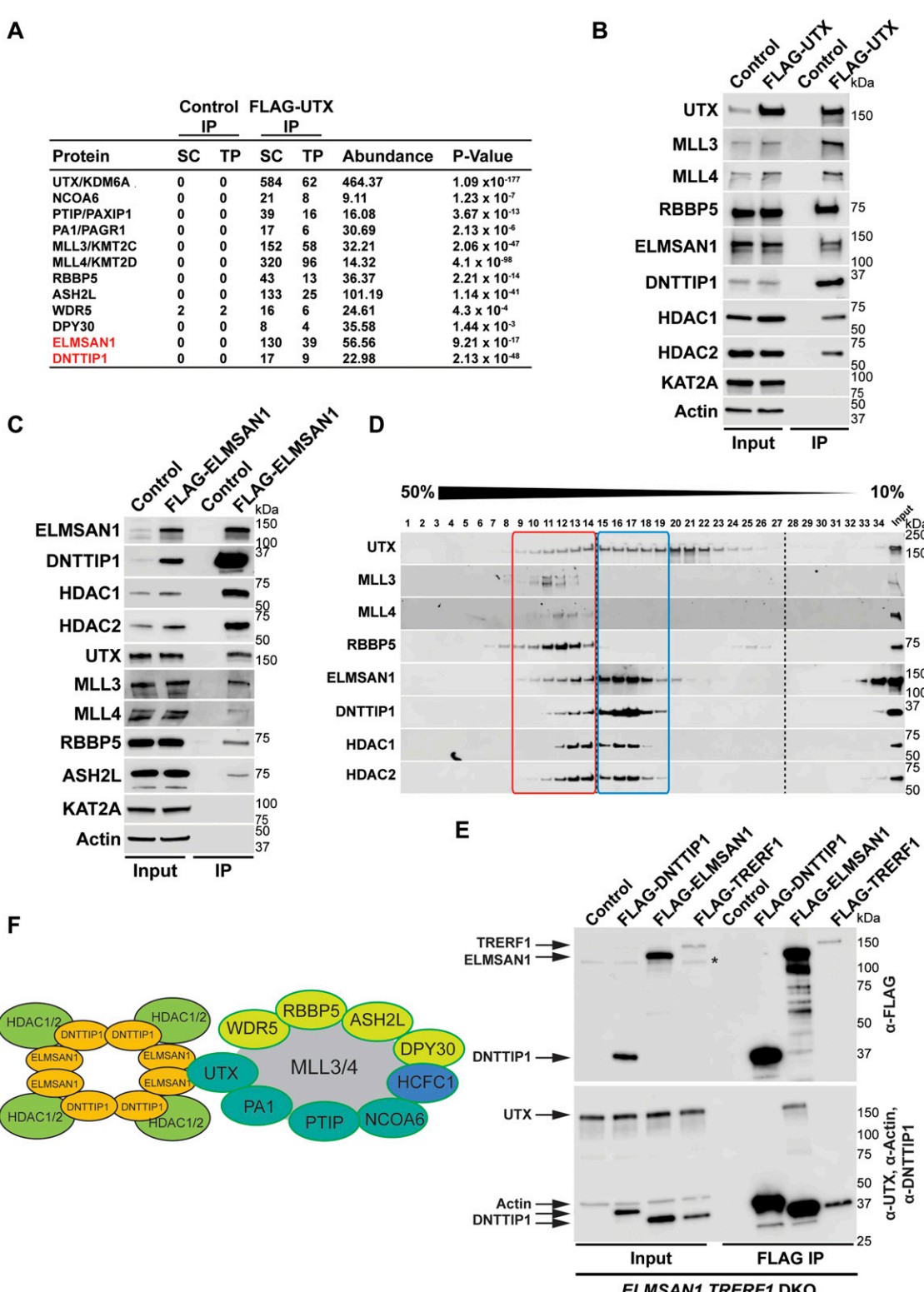

**Figure 1. The MLL3/4 complexes associate with the mitotic deacetylase complex (MiDAC).**

**(A)** FLAG-UTX immunoprecipitation (IP) followed by mass spectrometry (MS) identifies all known subunits of the MLL3/4 complexes along with ELMSAN1, and DNTTIP1. SC, spectral counts; TP, peptide counts; abundance = SC x 50 (kD)/protein size (kD). **(B)** Western blot (WB) of FLAG-UTX IP from HEK293 cells confirming interaction of UTX with ELMSAN1, DNTTIP1, HDAC1, and HDAC2. UTX interacts with the H3K4 methyltransferases MLL3 and MLL4, RBBP5 a core component of the MLL3/4 complexes, ELMSAN1, DNTTIP1, HDAC1, and HDAC2, but does not interact with KAT2A a histone acetyltransferase. HEK293 cells with a FLAG-tag expressing plasmid were used as an IP control. Nuclear extracts were used as input. Actin was used as a loading control for the inputs. **(C)** WB of FLAG-ELMSAN1 IP from HEK293 cells confirming interaction of ELMSAN1

MiDAC and MLL3/4 complexes in more detail (Figs 1E and S3). In *ELMSAN1 TRERF1* DKO cells, DNTTIP1 protein is almost completely lost (Fig 1E). Interestingly, reexpression of either FLAG-ELMSAN1 or FLAG-TRERF1 in *ELMSAN1 TRERF1* DKO cells resulted in stabilization of DNTTIP1, suggesting that MiDAC scaffolding subunits need to be co-expressed to retain each other's stability (Fig 1E). Reexpression of FLAG-DNTTIP1, FLAG-ELMSAN1, or FLAG-TRERF1 in *ELMSAN1 TRERF1* DKO cells followed by FLAG IPs, showed that only IPs from FLAG-ELMSAN1 expressing cells were able to purify UTX, whereas FLAG-DNTTIP1 and FLAG-TRERF1 expressing cells could not (Fig 1E). It should be noted, however, that FLAG-TRERF1 levels achieved by overexpression were significantly lower than FLAG-DNTTIP1 and FLAG-ELMSAN1 levels and thus might have prevented the detection of a potential interaction between TRERF1 and UTX (Fig 1E). To test whether UTX was able to interact with MiDAC outside of the MLL3/4 complexes, we used HCT116 colorectal carcinoma cells which contain a homozygous frameshift mutation in *MLL3* (*MLL3−/− MLL4+/+*) (Watanabe et al, 2011). We used CRISPR/Cas9 genome editing to also obtain HCT116 cells that are wild-type (WT) for *MLL3* (*MLL3+/+ MLL4+/+*) or contain a promoter deletion within the *MLL4* locus (*MLL3−/− MLL4−/−*) (Fig S4A and B). Transient transfection of FLAG-UTX into these HCT116 cell lines followed by FLAG IPs showed that UTX interaction with the MiDAC members DNTTIP1, ELMSAN1 and HDAC1 did not depend on MLL3/4 or core subunits of the MLL3/4 complexes such as RBBP5 and WDR5 suggesting a direct interaction between UTX and MiDAC (Fig S4C). In conclusion, our data imply that UTX functions as a bridging factor within the MLL3/4 complexes to mediate their association with MiDAC via its scaffolding subunit ELMSAN (Fig 1F).

## MiDAC is a genome-wide negative regulator of H4K20ac

We have previously shown by WB that a loss of MiDAC function in mESCs results in a bulk increase of H4K20ac (Mondal et al, 2020). To confirm these findings by an antibody-independent more quantitative method, we applied LC–MS/MS to measure the acetylation states of the N-terminal tails of H3 and H4 in WT and *Dnttip1* KO mESCs (Fig 2A). H4K20ac was the histone acetylation mark that displayed by far the strongest increase in *Dnttip1* KO compared with WT mESCs (Fig 2A–C). Except for H3K27ac and H4K16ac which both showed a decrease in acetylation across two independent *Dnttip1* KO clones, all other acetylation marks were not or only mildly affected (Fig 2A and B). Because H4K20ac was the mark with the strongest acetylation increase and its role in transcription regulation or other processes has not been studied to date we chose to focus on H4K20ac. To investigate site-specific changes in H4K20ac profiles genome-wide, we used ChIP-seq to profile H4K20ac in WT and *Dnttip1* KO mESCs. In addition, ChIP-seq was also carried out for H3K4me1/2/3, H3K27ac, and H3K27me3, UTX, and MLL4 in the same cell lines. Reanalysis of our previously published DNTTIP1

ChIP-seq data set from WT and *Dnttip1* KO mESCs (Mondal et al, 2020) identified a total of 38,572 DNTTIP1 binding sites (FC < 0.5, FDR < 0.05) (Fig 3A). Consistent with the bulk increase in H4K20ac in *Dnttip1* KO mESCs (Fig 2) (Mondal et al, 2020), we observe higher levels of H4K20ac on most of the DNTTIP1-bound genomic regions in *Dnttip1* KO compared with WT mESCs (Fig 3A and B). We also detect an enrichment of DNTTIP1 at nearly all H4K20ac sites in WT mESCs and this enrichment of DNTTIP1 is lost in *Dnttip1* KO mESCs (Fig S5A). Furthermore, we also find that H4K20ac is increased at the majority of all H4K20ac sites in *Dnttip1* KO compared with WT mESCs (Fig S5B). We also detected a milder increase of H3K4me2 and H3K4me3 without considerable alterations of H3K4me1 on regions that are targeted by MiDAC (DNTTIP1-bound regions) in *Dnttip1* KO versus WT mESCs (Fig 3C). Fig 3D depicts two enhancer regions and the promoter of the *Spry4* locus as an individual example (Fig 3D, red boxes). All three regulatory elements are bound by MiDAC (DNTTIP1) in WT mESCs and display a strong increase of H4K20ac upon *Dnttip1* KO, which is accompanied by higher enrichment of H3K4me2/me3 (Fig 3D, red boxes). In addition, we also observe a concomitant increase of H3K27ac and a tendency towards lower enrichment of the repressive H3K27me3 mark, whereas H3K4me1 is not significantly changed (Fig 3D, red boxes). In summary, our findings indicate that MiDAC constitutes a genome-wide negative regulator of H4K20ac by inhibiting the accumulation of H4K20ac on promoters and putative enhancers.

## MiDAC antagonizes the function of the MLL3/4 complexes on chromatin

To further dissect the functional relationship between MiDAC and the MLL3/4 complexes, we wanted to know how the loss of MiDAC function would affect chromatin occupancy of members of the MLL3/4 complexes. We found that a much higher fraction of UTX-bound or MLL4-bound regions displayed increased rather than decreased binding of UTX or MLL4 in *Dnttip1* KO compared with WT mESCs (Fig 4A). In total, we identified 4,826 peaks with higher UTX and 26,084 peaks with higher MLL4 occupancy in *Dnttip1* KO mESCs (both FC > 2) (Fig 4A). Interestingly, the DNTTIP1 binding sites identified in Fig 3C also displayed increased UTX and MLL4 occupancy in *Dnttip1* KO mESCs. Occupancy of UTX and MLL4 on putative enhancers and the promoter of the *Spry4* locus was also elevated in *Dnttip1* KO compared with WT mESCs (Fig 3D, red boxes). In addition, regions with up-regulated UTX binding (FC > 2) in *Dnttip1* KO mESCs also showed an increase in MLL4, and a tendency towards increased H4K20ac, H3K4me2, and H3K27ac enrichment (Fig S6A). Similarly, regions with up-regulated MLL4 binding (FC > 2) in *Dnttip1* KO mESCs displayed increased UTX, and a tendency towards increased H4K20ac, H3K4me2, and H3K27ac enrichment (Fig S6B). To assess the relationship between MiDAC and the MLL3/4 complexes more stringently, we subsequently only focused on the 1,283

with the MiDAC components DNTTIP1, HDAC1, HDAC2, and members of the MLL3/4 complexes including UTX, the H3K4 methyltransferases MLL3 and MLL4, and the two core components RBBP5 and ASH2L. KAT2A was used as a negative control. HEK293 cells with a FLAG-tag expressing plasmid were used as an IP control. Nuclear extracts were used as input. Actin was used as a loading control for the inputs. **(D)** Glycerol gradient sedimentation after FLAG-UTX IP reveals co-fractionation of the MiDAC subunits ELMSAN1, DNTTIP1, HDAC1, and HDAC2 with components of the MLL3/4 complexes (red box). Each antibody panel was assembled from three separate WBs with the assembly point marked by a black dashed line. **(E)** WB of FLAG IP from *ELMSAN1 TRERF1* double knockout (*ELMSAN1 TRERF1* DKO) HEK293 cells expressing FLAG-DNTTIP1, FLAG-ELMSAN1, or FLAG-TRERF1. Among the MiDAC subunits, only ELMSAN1 interacts with UTX. Reexpression of FLAG-ELMSAN1 or FLAG-TRERF1 in *ELMSAN1 TRERF1* DKO cells results in restoration of DNTTIP1 levels. * marks a nonspecific band. **(F)** Diagram showing that ELMSAN1 and UTX form the nexus between MiDAC and the MLL3/4 complexes.

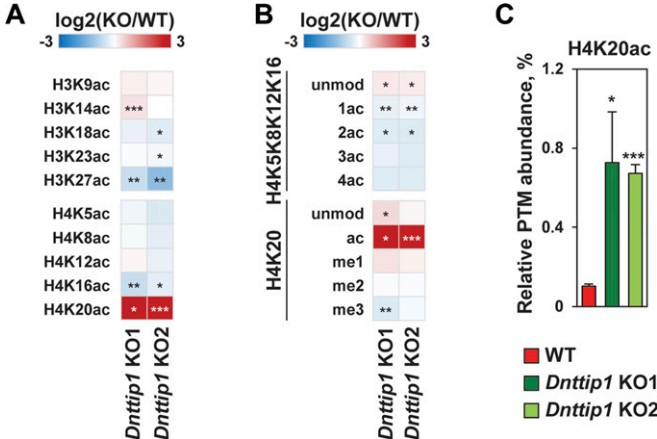

**Figure 2.   MiDAC is a global regulator of H4K20ac.**
**(A, B)** The heat maps show the log$_2$ fold change in the relative abundance of (A) various H3 and H4 acetylation sites or (B) H4 post translational modifications (PTMs) between WT and two different *Dnttip1* KO mESC clones as assessed by mass spectrometry. Single, double and triple asterisks indicate adjusted *P* < 0.05, adjusted *P* < 0.01 and adjusted *P* < 0.001, respectively (unpaired *t* test, *P*-values were corrected for multiple testing using the Benjamini–Hochberg method). Unmod = not acetylated on H4K5, H4K8, H4K12, H4K16, or H4K20. 1ac = one acetyl group detected on one lysine of H4K5, H4K8, H4K12, and H4K16. 2ac = two acetyl groups detected on two lysines of H4K5, H4K8, H4K12, and H4K16. 3ac = three acetyl groups detected on three lysines of H4K5, H4K8, H4K12, and H4K16. 4ac = acetyl groups detected on all lysines of H4K5, H4K8, H4K12, and H4K16. **(C)** Mass spectrometry quantification of the relative PTM abundance of H4K20ac expressed as a percentage (%) in WT and two different *Dnttip1* KO mESC clones. Data are the means ± SD of three independent experiments. Single and triple asterisks indicate adjusted *P* < 0.05 and adjusted *P* < 0.001, respectively (unpaired *t* test, *P*-values were corrected for multiple testing using the Benjamini–Hochberg method). H4K20ac levels are strongly increased in the absence of MiDAC function.

regions that displayed both increased UTX and MLL4 occupancy (FC > 2) in *Dnttip1* KO mESCs (Fig S6C). These UTX/MLL4 up-regulated regions also showed increased enrichment of H4K20ac, H3K4me2, and H3K27ac in *Dnttip1* KO versus WT mESCs (Fig 4B). Although the MLL3/4 complexes have been shown to catalyze both H3K4me1 and H3K4me2 on enhancers (Hu et al, 2013; Lee et al, 2013), we did not observe a significant change of H3K4me1 enrichment across the regions that showed both increased UTX and MLL4 occupancy in *Dnttip1* KO mESCs (Fig 4B). Similarly, no difference in H3K4me1 enrichment was detected on DNTTIP1-bound regions or individual promoter or putative enhancer elements of the *Spry4* locus in *Dnttip1* KO compared with WT mESCs (Fig 3C and D). Instead, in all cases, H3K4me2 was increased (Figs 3C and D and 4B). Based on this observation we also identified the regions with increased H3K4me2 enrichment (FC > 2, FDR < 0.05) in *Dnttip1* KO mESCs (2,396 regions) and observed higher enrichment of UTX, MLL4, H4K20ac, and H3K27ac without major changes in H3K4me1 on these sites in *Dnttip1* KO mESCs (Fig 4C). Interestingly, both promoter or putative enhancer regions with either increased H4K20ac or H3K4me2 enrichment or UTX/MLL4 occupancy tend to be associated with up-regulated gene expression in *Dnttip1* KO compared with WT mESCs (Fig 4D). Taken together, this suggests that MiDAC antagonizes the recruitment and/or activity of the MLL3/4 complexes at promoters and putative enhancers by regulating H4K20ac and H3K4me2 without significantly affecting H3K4me1.

## MiDAC and the MLL3/4 complexes oppose each other's function to balance a jointly regulated gene expression program

To better understand whether the relationship between MiDAC and the MLL3/4 complexes is based on reciprocity we performed ChIP-seq to profile DNTTIP1, H4K20ac, H3K4me1, H3K4me2, H3K4me3, H3K27ac, and H3K27me3 in WT and *Mll3/4* double knockout (DKO) mESCs (Fig S7) (Dorighi et al, 2017). Unexpectedly, most DNTTIP1-bound sites displayed decreased DNTTIP1 enrichment in *Mll3/4* DKO mESCs (Fig 5A). Altogether, we identified 10,462 sites with reduced DNTTIP1 occupancy (FC < 0.5, FDR < 0.05) in *Mll3/4* DKO compared with WT mESCs (Fig 5A). Interestingly, however, we observed that most regions with detectable H4K20ac showed a significant decrease of H4K20ac enrichment (5,513 sites) (FC < 0.5, FDR < 0.05) in *Mll3/4* DKO mESCs (Fig 5B). Likewise, this reduction in H4K20ac along with a reduction in H3K4me1, H3K4me2, H3K4me3, and H3K27ac was also detected at the 10,462 regions with reduced DNTTIP1 occupancy in *Mll3/4* DKO mESCs (Fig 5C). H3K27me3 seems to inversely correlate with the aforementioned histone marks showing an increase both at sites with decreased DNTTIP1 occupancy and regions with lower H4K20ac enrichment in *Mll3/4* DKO mESCs (Fig 5C). Thus, the detected changes in histone modification patterns between *Mll3/4* DKO and *Dnttip1* KO mESCs particularly as it pertains to H4K20ac, H3K4me2, H3K27ac, and H3K27me3 anti-correlate at regions that are co-regulated by MiDAC and the MLL3/4 complexes (Figs 3C and D, 4B and C, and 5C). This general finding is also obvious at putative enhancer regions and the promoter of the *Spry4* gene locus where we found H4K20ac and H3K27ac to be increased in *Dnttip1* KO mESCs but reduced in *Mll3/4* DKO mESCs, whereas H3K27me3 was reduced in *Dnttip1* KO mESCs but elevated in *Mll3/4* DKO mESCs (Fig 5D). This anticorrelative behavior as exemplified on the *Spry4* locus at the level of specific histone marks (Fig 5D) also aligned well with the gene expression changes we observed in *Dnttip1* KO compared with *Mll3/4* DKO mESCs. For example, promoters or putative enhancers with decreased enrichment of H4K20ac in *Mll3/4* DKO mESCs were preferentially associated with transcriptional repression (Fig S8), whereas promoters and putative enhancers with increased H4K20ac in *Dnttip1* KO mESCs displayed a significantly enhanced tendency for increased gene expression (Fig 4D). Furthermore, *Spry4* transcription was increased in *Dnttip1* KO but reduced in *Mll3/4* DKO mESCs (Figs 6A and S9). By transcriptome-wide analysis, we found a similar trend. Genes that were up-regulated in the absence of MiDAC function showed decreased expression in *Mll3/4* DKO mESCs, whereas down-regulated genes in *Dnttip1* KO mESCs displayed increased expression in Mll3/4 DKO mESCs (Fig 6B). Overall, this suggests that the MLL3/4 complexes and MiDAC act antagonistically as genome-wide regulators of H4K20ac to transcriptionally balance a jointly regulated gene expression program.

## Discussion

MiDAC is the least studied class I histone deacetylase complex and its important physiological role in neurogenesis and embryonic development has only very recently been described (Mondal et al,

**Figure 3. MiDAC is a genome-wide negative regulator of H4K20ac.**
**(A, B)** Volcano plots (left) and heat maps (right) showing the genome-wide loss of DNTTIP1 (A) and genome-wide increase of H4K20ac (B) in *Dnttip1* KO versus WT mouse embryonic stem cells (mESCs). Heat maps in (A) and (B) are centered on 38,572 identified DNTTIP1 peaks by comparing DNTTIP1 occupancy between WT and *Dnttip1* KO mESCs (FC > 2). **(C)** Heat maps centered on the 38,572 DNTTIP1 peaks identified in WT compared with *Dnttip1* KO mESCs in (A). Occupancy of H3K4me1, H3K4me2, H3K4me3, H3K27ac, UTX, and MLL4 in WT and *Dnttip1* KO mESCs is displayed. **(D)** Genome browser tracks depicting the ChIP-seq profiles of DNTTIP1, H4K20ac, H3K4me1, H3K4me2, H3K4me3, H3K27ac, H3K27me3, UTX, and MLL4 in WT and *Dnttip1* KO mESCs at the promoter and two enhancer regions of the *Spry*4 locus.

2020; Turnbull et al, 2020). How MiDAC functionally intersects with other epigenetic regulators to elicit changes in chromatin state and effect transcriptional responses is currently unknown. Here, we report for the first time a functional link between MiDAC and the MLL3/4 complexes. We show that UTX, a complex-specific subunit of the MLL3/4 complexes, acts as an important hub to ensure the

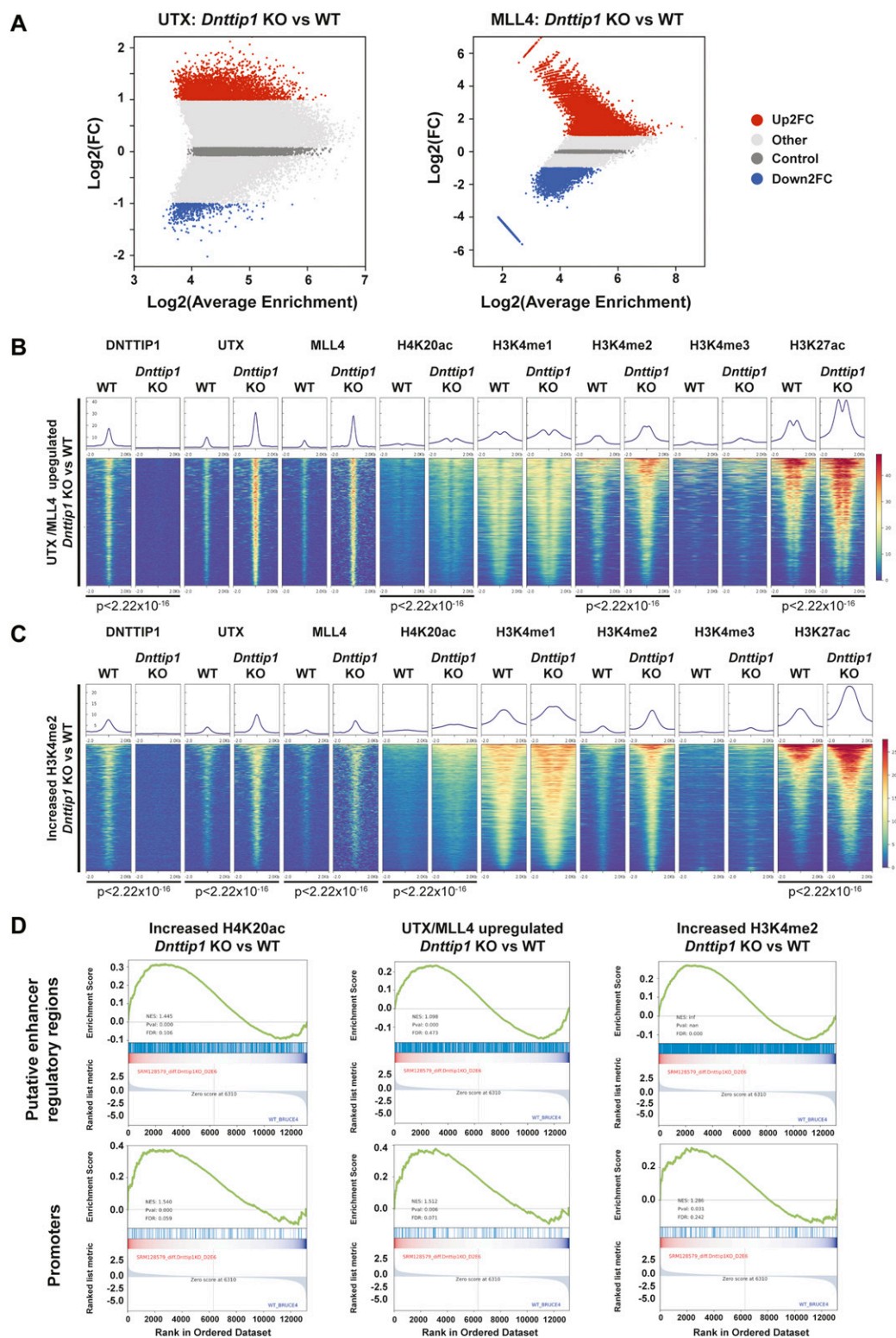

**Figure 4. MiDAC negatively regulates the occupancy of the MLL3/4 complexes at regulatory elements genome-wide.**
**(A)** MA plots displaying the log$_2$ fold change of UTX (left) and MLL4 (right) occupancy in *Dnttip1* KO compared with WT mESCs. A significantly higher number of UTX and MLL4 up-regulated peaks (FC > 2) than down-regulated peaks (FC > 2) is detected in *Dnttip1* KO versus WT mESCs. **(B, C)** Heat maps centered on (B) 1,283 UTX/MLL4 up-regulated peaks (both FC > 2) or (C) 2,396 regions with up-regulated H3K4me2 (FC > 2) in *Dnttip1* KO compared with WT mESCs. Occupancy of DNTTIP1, UTX, MLL4, H4K20ac, H3K4me1, H3K4me2, H3K4me3, and H3K27ac in WT and *Dnttip1* KO mESCs is displayed. The mean signal of each ChIP-seq peak was determined, and a two-sided Wilcoxon rank sum test was performed between WT and *Dnttip1* KO mESCs for each histone mark and protein ChIP sample for which a significant change is claimed. **(D)** Gene set enrichment analysis showing transcriptional up-regulation of genes associated with regions of increased H4K20ac (left), UTX/MLL4 up-regulated peaks (middle), and regions of increased H3K4me2 (right) in *Dnttip1* KO versus WT mESCs.

OFF

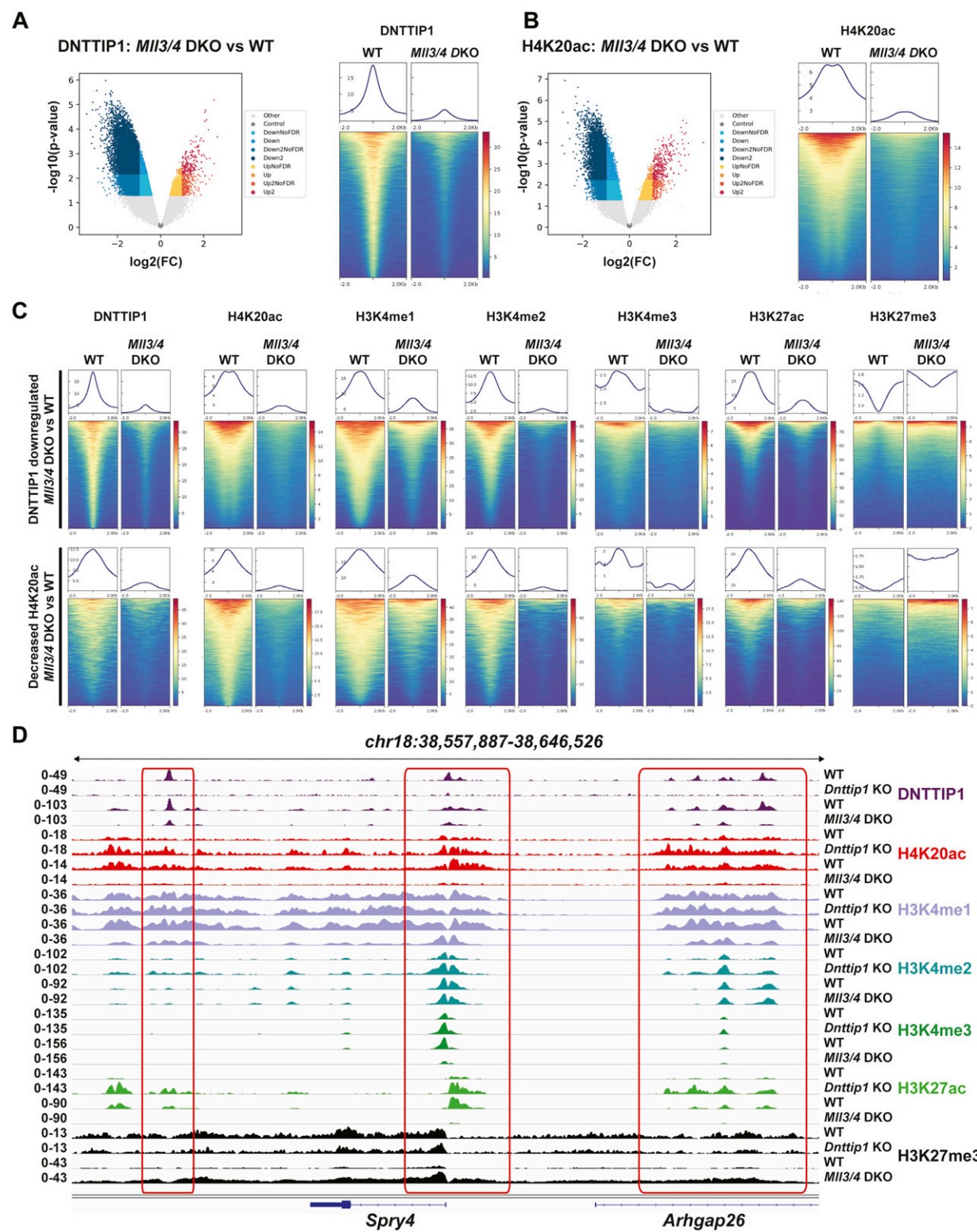

**Figure 5. MiDAC and the MLL3/4 complexes act antagonistically as genome-wide regulators of H4K20ac.**
**(A, B)** Volcano plots (left) and heat maps (right) showing the genome-wide reduction of DNTTIP1 (A) and genome-wide decrease of H4K20ac (B) in *Mll3/4* DKO compared with WT mESCs. Heat maps in (A) and (B) are centered on 10,462 DNTTIP1 peaks with a FC > 2 in *Mll3/4* DKO versus WT mESCs. **(C)** Heat maps centered either on 10,462 DNTTIP1 down-regulated peaks (FC > 2) (upper) or on 5,513 regions with decreased H4K20ac (FC > 2) (lower) in *Mll3/4* DKO compared with WT mESCs. Occupancy of DNTTIP1, H4K20ac, H3K4me1, H3K4me2, H3K4me3, H3K27ac, and H3K27me3 in WT and *Mll3/4* DKO mESCs is displayed. **(D)** Genome browser tracks depicting the ChIP-seq profiles of DNTTIP1, H4K20ac, H3K4me1, H3K4me2, H3K4me3, H3K27ac, and H3K27me3 in WT and *Dnttip1* KO mESCs, and WT and *Mll3/4* DKO mESCs at the promoter and two enhancer regions of the *Spry4* locus.

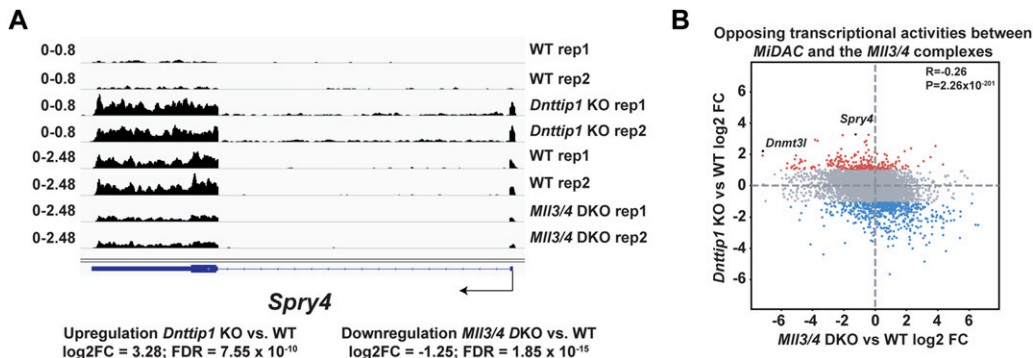

**Figure 6. A specific gene expression program is controlled by the opposing functions of MiDAC and the MLL3/4 complexes.**
**(A)** Genome browser tracks showing the RNA-seq profile of the *Spry4* gene in WT and *Dnttip1* KO mESCs, and WT and *Mll3/4* DKO mESCs. Two replicates are displayed for each genotype. **(B)** Scatter plot depicting the log$_2$ fold change of all expressed genes in *Dnttip1* KO versus WT mESCs (y-axis) and *Mll3/4* DKO versus WT mESCs (x-axis). A significant anticorrelation of differentially expressed genes that are transcriptionally controlled by both MiDAC and the MLL3/4 complexes is observed.

physical association with MiDAC via its scaffolding subunit ELMSAN1 (Fig 1). Our FLAG-affinity UTX purification identified two different MiDAC-containing UTX complexes, one in which UTX is a part of the MLL3/4 complexes (Fig 1D, red box) and another UTX-MiDAC complex that does not contain subunits of the MLL3/4 complexes (Fig 1D, blue box). These findings are puzzling as they might suggest that UTX can form a complex with MiDAC independent of the MLL3/4 complexes. However, the association of UTX with MiDAC outside the MLL3/4 complexes (Fig 1D, blue box) could be the result of UTX overexpression. The overabundance of UTX due to over-expression might not permit incorporation of a significant portion of UTX into the MLL3/4 complexes because of the limited endog-enous amounts of other subunits of the MLL3/4 complexes (par-ticularly MLL3/4). Because UTX most likely constitutes the bridging factor between MiDAC and the MLL3/4 complexes, this UTX "sur-plus" (outside the MLL3/4 complexes) is still able to interact with MiDAC so that the UTX–MiDAC interaction independent of the MLL3/4 complexes (Fig 1D, blue box) would not represent a physiologically relevant complex. We favor the possibility that UTX associates with MiDAC only within the context of the MLL3/4 complexes because UTX is highly unstable in the absence of MLL3/4 (Figs S2, S4B, and S7). This dependency of UTX on MLL3/4 makes it likely that most if not all functions of UTX are carried out through the MLL3/4 complexes and that the observed interaction of overexpressed UTX with MiDAC outside the MLL3/4 complexes does not represent a physiologically relevant complex. Indeed, apart from mediating the interaction with MiDAC, UTX might generally function as a unique docking site within the MLL3/4 complexes to recruit other chromatin-associated protein complexes, including the previously reported TOP (TET2, OGT, and PROSER1) complex (Wang et al, 2022). Interestingly, *UTX* is often mutated in the same cancer types as *MLL3* and/or *MLL4* linking the function of the MLL3/4 complexes directly to carcinogenesis via UTX (Martinez-Jimenez et al, 2020). Thus, gaining mechanistic insight into the role of UTX-interacting chromatin-modifying complexes such as MiDAC and the TOP complex might open an avenue to specifically target these cancers. The results of our study implicate MiDAC and the MLL3/4 complexes as antagonistic co-regulators of a specific gene expression program (Fig 7). This relationship is clearly evident at the level of histone

modifications. We previously identified MiDAC as a major negative regulator of H4K20ac (Mondal et al, 2020) and here provide evidence that this function of MiDAC towards H4K20ac extends to many gene regulatory elements genome-wide (Fig 3). Although our data suggest that MiDAC directly deacetylates H4K20ac, we cannot exclude the possibility that the observed changes in H4K20ac are indirect and might be controlled by other histone deacetylase complexes or histone acetyltransferases whose catalytic activities and/or re-cruitment might be altered as a result of lost MiDAC function. Conversely, the MLL3/4 complexes function to promote H4K20ac deposition on many loci that are co-regulated by the MLL3/4 complexes and MiDAC (Fig 5). These findings imply that the degree of H4K20ac enrichment on promoters and putative enhancers might be used as an indicator to predict the transcriptional response of a gene expression program that is balanced by the opposing action of MiDAC and the MLL3/4 complexes. To date our understanding of H4K20ac is extremely limited. Thus, future work is required to further elucidate the role of H4K20ac in transcription and potentially other processes. One possible mechanism by which the MLL3/4 complexes might regulate H4K20ac could be through direct binding to acety-lated histones. For example, previous studies have shown that the PHD6 domain of MLL4 is able to recognize and bind H4K16ac (Zhang et al, 2019). Therefore, it is possible that other PHD domains or re-gions within MLL3 and/or MLL4 or domains/regions within UTX might recognize H4K20ac and thus protect H4K20ac from being deacety-lated by MiDAC. Alternatively, the MLL3/4 complexes might promote the activity or recruitment of a histone acetyltransferase that is able to target H4K20ac. CBP/p300 are histone acetyltransferases that have been known to collaborate with the MLL3/4 complexes in enhancer activation both in *Drosophila* and mammals and thus might be potential candidates in catalyzing H4K20ac (Tie et al, 2012; Wang et al, 2016, 2017; Lai et al, 2017).

Unexpectedly, we found that the decrease of H4K20ac on gene regulatory elements in *Mll3/4* DKO mESCs did not coincide with higher MiDAC occupancy on these loci but rather a decrease in MiDAC enrichment. As the protein levels of MiDAC components are unchanged in *Mll3/4* DKO mESCs (Fig S7), this suggests that the genomic localization of MiDAC is dependent on the MLL3/4 com-plexes and that in the absence of the MLL3/4 complexes, the

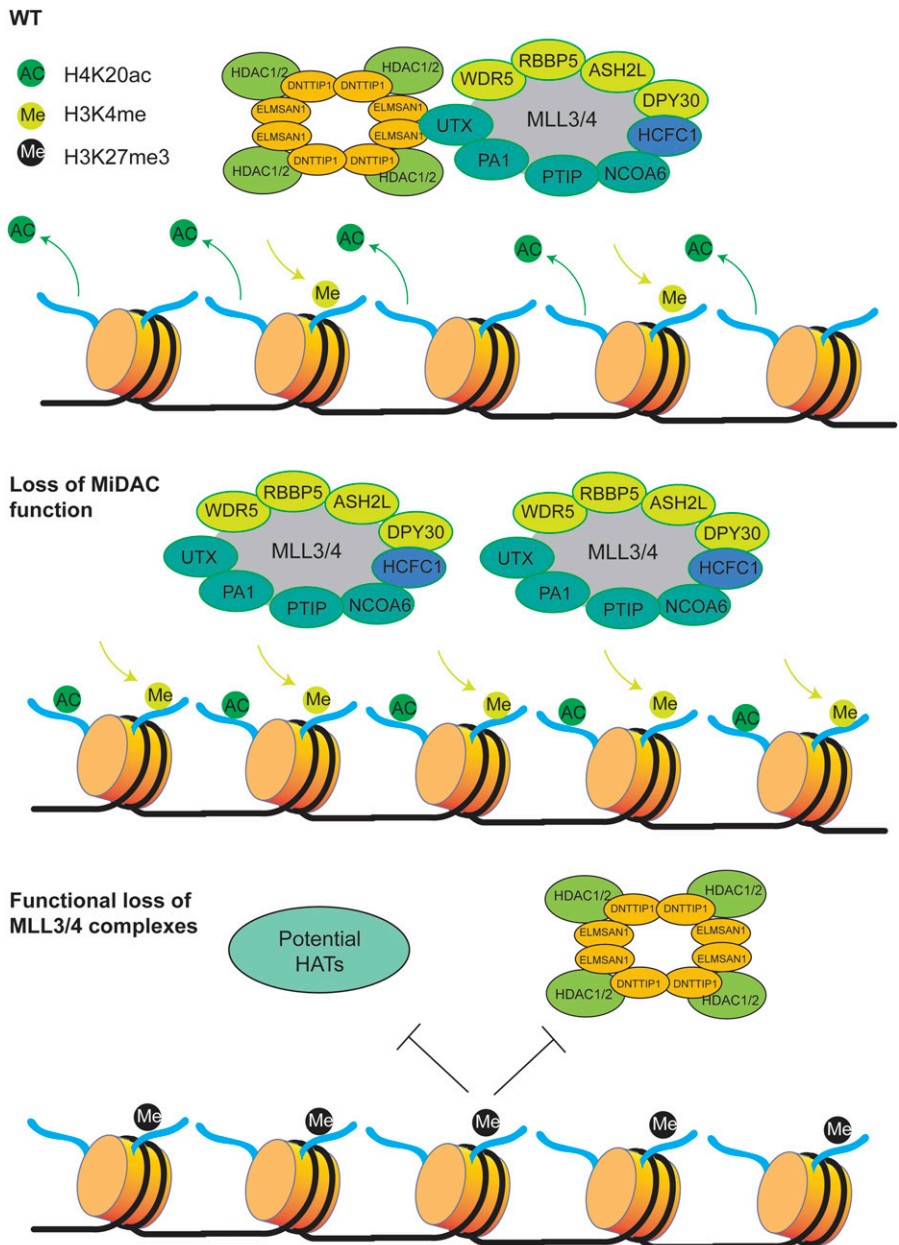

**Figure 7. Model describing the interplay between MiDAC and MLL3/4 complexes.**

Under WT conditions, MiDAC functionally intersects with the MLL3/4 complexes through the interaction between ELMSAN1 and UTX. The removal of H4K20ac and deposition of H3K4me2 are in a balanced state. Upon loss of MiDAC, H4K20ac increases genome-wide and by an unknown mechanism the MLL3/4 complexes' deposition also increases, thus increasing H3K4me2 on genomic elements that are co-regulated by MiDAC and the MLL3/4 complexes. This indicates that under normal conditions, MiDAC may play an inhibitory role towards the MLL3/4 complexes. Upon loss of the MLL3/4 complexes, MiDAC is reduced genome-wide which intriguingly does not result in an increase of the MiDAC substrate H4K20ac. This may be partially due to elevated H3K27me3 preventing the access of potential histone acetyltransferases (HATs) and MiDAC.

inhibitory role of MiDAC towards H4K20ac is suppressed. It is possible that the MLL3/4 complexes are directly involved in the recruitment of MiDAC to its target loci or that an unknown MiDAC recruiting factor is reduced in *Mll3/4* DKO mESCs. The decrease of H3K4me1 in *Mll3/4* DKO mESCs could also cause the reduction in MiDAC occupancy, as MiDAC deposition may be dependent on direct binding to H3K4me1. Interestingly, however, we observed higher enrichment of the repressive histone mark H3K27me3 at loci with reduced MiDAC occupancy or decreased H4K20ac enrichment in *Mll3/4* DKO mESCs. Thus, the formation of a repressive chromatin state (via H3K27me3) may prevent MiDAC and potential histone acetyltransferases that target H4K20 to access those regions. Furthermore, we cannot exclude that one or multiple H4K20

deacetylase(s) is/are activated and/or target H4K20ac more effectively in *Mll3/4* DKO mESCs. A potential model of the interplay between MiDAC and the MLL3/4 complexes is described in Fig 7. Besides H4K20ac, other histone marks such as H3K4me2 and to some degree H3K27ac are also antagonistically regulated by MiDAC and the MLL3/4 complexes (Figs 3–5). It has previously been shown that MLL3/4 can catalyze both H3K4me1 and H3K4me2 on enhancers (Hu et al, 2013; Lee et al, 2013). However, the increased occupancy of UTX and MLL4 we observed in *Dnttip1* KO mESCs on regulatory elements that are also targeted by MiDAC did not result in higher H3K4me1 enrichment, but selectively appeared to be confined to increased H3K4me2 and H3K27ac enrichment (Figs 3 and 4). One potential reason why we observed no significant

changes of H3K4me1 in *Dnttip1* KO mESCs may be due to an increased propensity of the MLL3/4 complexes to catalyze H3K4me2 rather than H3K4me1 in the presence of H4K20ac. Finally, our transcriptome-wide analysis clearly confirms the antagonistic relationship between MiDAC and the MLL3/4 complexes and aligns well with our observations on the level of histone modifications as discussed above (Fig 6). Future studies will be required to further interrogate the role of MiDAC and the MLL3/4 complexes in development and disease.

# Materials and Methods

## Cell lines

*Dnttip1* KO (clone 1-D1E6) and their wild-type control mESCs were previously described by our laboratory (Mondal et al, 2020). *Mll3/4* DKO, *Mll3/4* DCD, and their wild-type control mESCs were kindly provided by the Wysocka laboratory (Dorighi et al, 2017). All mESCs were cultured under chemically defined naïve culture conditions (2iL) in 0.1% gelatin-coated flasks or plates. *ELMSAN1 TRERF1* DKO HEK293 cells were generated from wild-type Flp-In T-REx HEK293 cells (R78007; Invitrogen) by the Center for Advanced Genome Engineering (CAGE) at St. Jude Children's Research Hospital using CRISPR/Cas9–mediated gene editing (Fig S3). HCT116 cells (*MLL3−/− MLL4+/+*) were obtained from ATCC. *MLL3+/+ MLL4+/+* HCT116 cells and *MLL3−/− MLL4−/−* HCT116 cells were generated from *MLL3−/− MLL4+/+* HCT116 cells by the CAGE at St. Jude Children's Research Hospital using CRISPR/Cas9–mediated gene editing (Fig S4). Isogenic tetracycline-inducible FLAG-UTX and FLAG control HEK293 cells were previously described by our laboratory (Wang et al, 2022). Isogenic tetracycline-inducible FLAG-DNTTIP1, FLAG-ELMSAN1, FLAG-TRERF1, and FLAG control HEK293 cells were generated by using Flp recombinase-mediated integration in *ELMSAN1 TRERF1* DKO HEK293 cells. HEK293 and HCT116 cells were cultured in Dulbecco's Modified Eagle Medium (11995065; Gibco) with 10% FBS (97068-085; VWR) and 1% penicillin/streptomycin (15140122; Gibco). mESC identity was authenticated by various methods including alkaline phosphatase staining and staining and cytometry analysis of the pluripotency marker FUT4 (SSEA-1). All HEK293 and HCT116 cell lines were authenticated using STR profiling and tested negative for mycoplasma contamination.

## Antibodies

### For Western blotting
Mouse α-Actin (Developmental Studies Hybridoma Bank, JLA20 supernatant) at 1:1,000; rabbit α-ASH2L (5019S; Cell Signaling Technology) at 1:2,000; rabbit α-DNTTIP1 (A304-048A; Bethyl Laboratories) at 1:2,000; rabbit α-ELMSAN1 (A303-157A; Bethyl Laboratories) at 1:5,000; rabbit α-ELMSAN1 (34421; Herz Lab) at 1:5,000; mouse α-FLAG (F3165; Sigma-Aldrich) at 1:2,000; rabbit α-HDAC1 (34589S; Cell Signaling Technology) at 1:2,000; rabbit α-HDAC2 (57156S; Cell Signaling Technology) at 1:2,000; rabbit α-KAT2A (3305S; Cell Signaling Technology) at 1:2,000; rabbit α-MLL3/KMT2C (#31865 and #31866 [both human aa 581–850]; Herz Lab) at 1:5,000; rabbit

α-MLL4/KMT2D (#31863 [human aa 1–181]) and #32757 [human aa 281–506]; Herz Lab at 1:5,000; rabbit α-RBBP5 (13171S; Cell Signaling Technology) at 1:2,000; rabbit α-TRERF1 (HPA051273; Sigma-Aldrich) at 1:2,000; rabbit α-UTX/KDM6A (33510S; Cell Signaling Technology) at 1:2,000; α-WDR5 (13105S; Cell Signaling Technology) at 1:2,000.

### For immunoprecipitations (IPs)
Rabbit α-DNTTIP1 (A304-048A; Bethyl Laboratories), 5 μg per IP; rabbit IgG (011-000-003; Jackson ImmunoResearch), 5 μg per IP; rabbit α-UTX/KDM6A (A302-374A; Bethyl Laboratories), 5 μg per IP.

### For ChIP-seq
Rabbit α-H3K4me1 (31-1046-00; RevMAb), 10 μg per ChIP; mouse α-H3K4me2 (39679; Active Motif), 10 μg per ChIP; rabbit α-H3K4me3 (31-1039-00; RevMAb), 10 μg per ChIP; rabbit α-H3K27ac (31-1056-00; RevMAb), 10 μg per ChIP; rabbit α-H3K27me3 (9733S; Cell Signaling Technology), 30 μl per ChIP; rabbit α-H4K20ac (31-1084-00; RevMAb), 10 μg per ChIP; rabbit α-DNTTIP1 (A304-048A; Bethyl Laboratories), 10 μg per ChIP; rabbit α-UTX/KDM6A (33510S; Cell Signaling Technology), 30 μl per ChIP; rabbit α-MLL4/KMT2D (#3; Ge Lab), 10 μg per ChIP.

## CRISPR/Cas9 gene editing

*ELMSAN1 TRERF1* DKO clones from Flp-In T-REx HEK293 and *MLL3+/+ MLL4+/+* clones and *MLL3−/− MLL4−/−* clones from HCT116 cells were generated using CRISPR/Cas9 technology. Briefly, 400,000 Flp-In T-REx HEK293 cells (R78007; Thermo Fisher Scientific) were transiently transfected with precomplexed RNPs consisting of 100 pmol of each chemically modified sgRNA (Synthego), 70 pmol of Cas9 protein (St. Jude Protein Production Core), and 200 ng of pMaxGFP (Lonza) via nucleofection (4D-Nucleofector X-unit; Lonza) using solution P3 and program CM130 in a small (20 μl) cuvette according to the manufacturer's recommended protocol. 5 d post transfection, cells were single cell sorted by FACS to enrich for GFP-positive (transfected) cells, clonally selected and verified for the desired targeted modification via targeted deep sequencing. Targeted amplicons were generated using gene specific primers with partial Illumina adapter overhangs and sequenced as previously described (Sentmanat et al, 2018). Briefly, cell pellets of ~10,000 cells were lysed and used to generate gene specific amplicons with partial Illumina adapters in PCR #1. Amplicons were indexed in PCR #2 and pooled. In addition, 10% PhiX Sequencing Control V3 (Illumina) was added to the pooled amplicon library before running the sample on a Miseq Sequencer System (Illumina) to generate paired 2 × 250-bp reads. Samples were demultiplexed using the index sequences, fastq files were generated, and NGS analysis was performed using CRIS.py (Connelly & Pruett-Miller, 2019). Two clones were initially identified, and one was used for further characterization as it pertains to this article. Editing construct sequences and relevant primers are listed in Table S2.

## Immunoprecipitation (IP)

Large scale IPs from ten 150-mm plates and small scale IPs from one 150-mm plate were carried out as previously reported (Wang et al, 2022).

## Glycerol gradient fractionation

Glycerol gradient fractionation was conducted as previously described (Wang et al, 2022). 34 fractions of ~325 µl each were collected and analyzed by WB for UTX, MLL3, MLL4, RBBP5, ELMSAN1, DNTTIP1, HDAC1, and HDAC2.

## ChIP-seq

ChIP-seq was performed as previously described (Mondal et al, 2020; Wang et al, 2022).

## MS

### Protein identification by liquid chromatography coupled with tandem MS

**Sample preparation** Protein samples were briefly run into a 4–20% PAGE gradient gel as described in a previously published protocol (Xu et al, 2009). The gel bands were destained, reduced with DTT, alkylated by iodoacetamide (IAA), washed, dried down, and rehydrated with a buffer containing trypsin. After overnight proteolysis at 37°C, peptides were extracted, dried down in a speed vacuum and reconstituted in 5% formic acid.

**MS** The peptide mixture from each gel band was separated on a nanoscale capillary reverse phase C18 column (75 µm id, 10 cm) by a HPLC system (Thermo EASY-nLC 1000). Buffer A was 0.2% formic acid and Buffer B was 0.2% formic acid in 70% acetonitrile. The peptides were eluted by increasing Buffer B from 12% to 70% over a 60–90-min gradient. The peptides were ionized by electrospray ionization and detected by a Thermo LTQ Orbitrap Elite mass spectrometer. The mass spectrometer was operated in data-dependent mode. For each duty cycle, a high-resolution survey scan in the Orbitrap and 20 low-resolution MS/MS scans were acquired in the ion trap.

**Database search and analysis** The MS data were searched against the human UniProt database using Sequest (version 28, rev. 12) (Eng et al, 1994). The database was concatenated with a reversed decoy database for evaluating false discovery rate (FDR) (Peng et al, 2003). Mass tolerance of 15 ppm for precursor ions and 0.5 D for product ions were used. Two missed cleavages with a maximum of three modifications were allowed and assignment of b, and y ions were used for identification. Carbamidomethylation of cysteine (+57.02146 D) for static modification and oxidation of Methionine (+15.99492 D) for dynamic modification were considered. Mass accuracy and matching score filters were used for MS/MS spectra to reduce the protein FDR to <1%. Spectral counts of each protein may reflect their relative abundance in the samples after normalizing for protein molecular weight. The spectral counts between the samples for a given protein were used to calculate the $P$-value which was derived by the G-test (Bai et al, 2013).

### Relative quantification of histone post translational modification abundances using LC–MS/MS

**Sample preparation** Histones were acid extracted as described previously (Shechter et al, 2007). In brief, mESCs were lysed in 10X cell pellet volumes of ice-cold hypotonic lysis buffer (15 mM Tris–HCl [pH 7.5], 60 mM KCl, 11 mM CaCl$_2$, 5 mM NaCl, 5 mM MgCl$_2$, 250 mM sucrose, 1 mM DTT, and 10 mM sodium butyrate) supplemented with 0.1% NP-40 on ice for 5 min. Nuclei were pelleted by centrifugation (1,000$g$, 2 min, 4°C) and washed twice in ice-cold hypotonic lysis buffer w/o NP-40. Nuclei were resuspended in 5X nuclei pellet volumes of ice-cold 0.2 M sulfuric acid and mixed on a rotation wheel for 120 min at 4°C. Insolubilized nuclear debris was pelleted by centrifugation (16,000$g$, 10 min, 4°C). Supernatant was transferred to a fresh low-protein binding Eppendorf tube and histone proteins were precipitated by adding ice-cold trichloroacetic acid (TCA) to the final concentration of 20% (vol/vol) followed by a 60-min incubation on ice. Precipitated histone proteins were pelleted by centrifugation (16,000$g$, 10 min, 4°C), washed three times with acetone (−20°C), and resuspended in MS-grade water.

**MS** Extracted histones were prepared for LC–MS/MS analysis using the hybrid chemical derivatization method as described previously (Maile et al, 2015). In brief, 4 µg aliquots of purified histones were diluted with MS grade water to a total volume of 18 µl and buffered to pH 8.5 by addition of 2 µl of 1 M triethylammonium bicarbonate buffer (TEAB). Propionic anhydride was mixed with MS grade water in a ratio of 1:100 and 2 µl of the anhydride-mixture was added immediately to the histone sample, with vortexing, and the resulting mixture was incubated for 5 min at room temperature. The reaction was quenched by adding 2 µl of 80 mM hydroxylamine followed by a 20-min incubation at room temperature. Tryptic digestion was performed overnight with 0.5 µg trypsin per sample at 37°C. A 1% vol/vol solution of phenyl isocyanate (PIC) in acetonitrile was freshly prepared and 6 µl added to each sample and incubated for 60 min at 37°C. Samples were acidified by adding TFA to a final concentration of 1%. Peptides were de-salted with C18 spin columns (Pierce) following the manufacturer's protocol. Peptides were eluted from C18 spin columns with 70% acetonitrile, partially dried in a speedvac and resuspended in 30 µl 0.1% TFA.

The resulting peptide mixtures were analyzed using nano-flow liquid chromatography tandem mass spectrometry (LC–MS/MS) on a Q-Exactive HF mass spectrometer coupled to an Ultimate 3000 nano-UPLC (Ultimate 3000; Dionex) in data-dependent acquisition mode. A ~200 ng peptide aliquot was used per one sample per one injection. Peptides were loaded automatically on a trap column (300 µm inner diameter ×5 mm, Acclaim PepMap100 C18, 5 µm, 100 Å; LC Packings) before C18 reversed-phase chromatography on the analytical column (nanoEase MZ HSS T3 Column, 100 Å, 1.8 µm, 75 µm × 250 mm; Waters). Peptides were separated at a flowrate of 0.250 µl per minute by a linear gradient from 1% buffer B (0.1% [vol/vol] formic acid, 98% [vol/vol] acetonitrile) to 25% buffer B over 40 min followed by a linear gradient to 40% B in 20 min, then to 85% B in 5 min. After 5 min at 85% buffer B, the gradient was reduced to 1% buffer B over 2 min and then allowed to equilibrate for 8 min. Full mass range spectra were at 60,000 resolution (at m/z 400), and product ions spectra were collected in a "top 15" data-dependent scan cycle at 15,000 resolution.

**Data analysis** RAW MS data were analyzed using EpiProfile 2.0 software (Yuan et al, 2018). The reported relative abundances of histone modifications were validated manually using an open-source Skyline software.

## ChIP-seq data processing

Following the procedure described before (Wang et al, 2022), raw sequencing reads were pre-processed with the Trim-Galore tool (v0.4.4, https://www.bioinformatics.babraham.ac.uk/projects/trim_galore/) (Krueger et al, 2012) and cutadapt (DOI: 10.14806/ej.17.1.200), to remove low quality reads, remove potential adapters and quality trim reads' 3' ends. Quality score cutoff was set to Q20. Next, the remaining reads were mapped to the mouse reference genome (mm10) with bwa aln, followed by bwa samse (Li & Durbin, 2009) (v0.7.12-r1039) with -K flag set to 10,000,000. The output was then converted to binary alignment map format with SAMtools (Li et al, 2009) (v1.2). Next, the bamsormadup tool from biobambam2 (v2.0.87, DOI: 10.1186/1751-0473-9-13) was used to identify duplicated reads, and the SPP tool (Kharchenko et al, 2008) (v1.11) was used to conduct the Cross-Correlation analysis and estimate the fragment size. Subsequently, SAMtools was used again to extract uniquely mapped reads, and bedtools (Quinlan & Hall, 2010) (v2.24.0) was then used to extend the reads with the previously estimated fragment size. The intermediate files containing the extended fragments, were converted to bigwig track files by University of California, Santa Cruz (UCSC) tools (Kuhn et al, 2013) (v4), and signal intensity was corrected for sequencing depth, normalizing them to 15 million uniquely mapped non-duplicated fragments. For visualization purposes, for the experiments that per condition consisted of more than one replicate, the bigwig files from individual replicates were merged calculating average per bin signal between replicates. Subsequently, MACS2 (Zhang et al, 2008) was used to call peaks in narrow mode, with –nomodel -q 0.05 flags (high confidence peaks). Separately, narrow peaks were also called with more relaxed criteria, setting the -q flag to 0.5, which are here referred to as FDR50 peaks. Next, for experiments with more than one replicate, reproducible peaks (e.g., for H4K20ac in *Dnttip1* KO mESCs) were identified as those with overlapping FDR50 peaks present in all replicates at a given genomic region. Otherwise, for ChIP-seq targets without replicates, only the high confidence peaks were considered. Finally, the reproducible peaks from the same immunoprecipitation target (e.g., H4K20ac) were merged into the collection of reference peaks, here further referred to as all reproducible peaks.

## Differential binding peak identification

To perform statistical testing between experimental groups, the number of fragments for each reference peak was counted with intersect command from pybedtools (Quinlan & Hall, 2010; Dale et al, 2011) (v0.8.1). Next, the number of raw fragments mapping per peak was converted to FPKM units (Fragments Per Kilo base per Million mapped reads); and then TMM (trimmed mean of M-values) from edgeR (Robinson et al, 2010), followed by the limma-voom approach (Ritchie et al, 2015) was used to assess the significance of differential peak binding. For the contrasts, for which based on previously published independent work (Dorighi et al, 2017; Mondal et al, 2020), a genome-wide change (either gain or loss) was expected, and for which no spike-in was available, an additional pre-processing step was introduced to calculate the scaling factors. The contrasts for this step included DNTTIP1, H4K20ac, UTX, and MLL4 in *Dnttip1* KO mESCs

and H4K20ac, H3K4me1 and DNTTIP1 in *Mll3/4* DKO mESCs. Those scaling factors were computed by first calculating the median of single base-pair resolution enrichment signal from the bigwig track files, corrected previously for sequencing depth (see above), which was accomplished with the pybigwig tool (available online at: https://github.com/deeptools/pyBigWig), over all reproducible peaks. Next, the median signal from each sample was multiplied by 1/1,000,000, and converted to integer, which value imitated the spike-in read counts for the purpose of the scaling factor calculation and is further referred to as pseudo-spike reads. Next, similar to the approach used by the authors of DESeq2 and edgeR (Robinson et al, 2010; Love et al, 2014) the scaling factors were calculated normalizing individual pseudo-spike reads to the maximum pseudo-spike read counts across samples, and then dividing these values by their geometric mean. Scaling factors calculated this way, were further supplied to the *norm.factors* parameter of the DGElist function of edgeR for differential peak calling. For the contrasts with replicates, the region was considered as differentially binding, when the FDR was lower than 0.05 and the $\log_2$(fold-change) > 1, for increased binding, or $\log_2$(fold-change) < –1 for decreased binding. For some experiments without available biological replicates (including UTX and MLL4 for *Dnttip1* KO mESCs and H3K4me1, H3K4me3, and H3K27Ac for *Mll3/4* DKO mESCs), the identification of differentially bound regions was based on the fold-change value, using a $\log_2$(fold-change) > 1 threshold to identify increased binding and $\log_2$(fold-change) < –1 for decreased binding.

## Annotation of genomic regions

Following the approach used previously (Wang et al, 2022), genomic regions were assigned to their genomic contexts with an in-house script based on pybedtools (Dale et al, 2011) (v0.8.1), such that each region could only be assigned to one feature. For this purpose, genomic regions were successively overlapped with predefined genomic contexts in the following prioritization order: (1) Promoter.Up: region up to 2 kbp upstream from TSS; (2) Promoter.Down: region up to 2 kbp downstream from TSS; (3) Exons; (4) Introns; (5) TES: transcription end sites; (6) 5' Distal: region up to 50 kbp upstream from TSS, excluding promoter region; (7) 3' Distal: region up to 50 kbp downstream from TSS, excluding promoter region; (8) Intergenic. The reference annotation for TSS, and all subsequent genomic contexts, was based on the Gencode vM14 (Frankish et al, 2019) reference annotation and included all isoforms. In parallel, the genomic regions were annotated with genes, via putative promoter-related association. For that purpose, genomic regions were overlapped with promoter regions with bedtools (Quinlan & Hall, 2010) (v2.24.0); one region could be assigned to multiple genes. The promoter region was defined as TSS ± 2 kbp. Next, regions not assigned to any gene as promoter-associated, were assigned to a gene as putative enhancer-related regions, if their distance to the gene's TSS was within a threshold of ± 50 kbp, excluding the promoter region.

## RNA-seq data processing

Sequenced RNA-seq reads were quality-filtered using TrimGalore (https://www.bioinformatics.babraham.ac.uk/projects/trim_galore/),

and then aligned to the mouse reference genome (mm10) using STAR (Dobin et al, 2013). Next, RSEM was used to quantify read counts per gene (Dobin et al, 2013). Subsequently, differentially expressed genes were identified using the limma-voom approach (Law et al, 2014; Ritchie et al, 2015), as previously described (Wang et al, 2022).

### Gene set enrichment analysis (GSEA)

GSEA was conducted with GSEApy (Mootha et al, 2003; Dale et al, 2011) (v0.10.4, available on-line at https://gseapy.readthedocs.io/en/latest/), using the pre-ranked list of genes, where the per gene metric was an equivalent of the $\log_2$(fold-change) values, derived from differential gene expression analyses. GSEApy's prerank function was used with standard parameters, except –min-size and –max-size flags, whose values were set to 5 and 5,000, respectively. The GSEA analysis was run with the collection of gene sets from the KEGG database (Kanehisa & Goto, 2000), downloaded from the Enrichr portal (Kuleshov et al, 2016). The abovementioned gene sets collection was additionally expanded with in-house gene sets, representing genes annotated with various genomic regions, for example, regions with increased H4K20ac, or genomic regions displaying both an increase in UTX and MLL4 occupancy. Following previous approaches (Subramanian et al, 2005) (Bayá et al, 2007), we considered the gene sets that are significantly associated with positive or negative phenotype as the ones whose FDR was lower than 25% and whose *P*-value was lower than 0.05.

## Data Availability

RNA-seq data from WT and *Dnttip1* KO mESCs were obtained from GSE131062. RNA-seq data from WT and *Mll3/4* DKO mESCs were obtained from GSE98063. DNTTIP1 ChIP-seq data in WT and *Dnttip1* KO mESCs were used from GSE131062. The accession number for the remaining ChIP-seq datasets reported in this article is GEO: GSE190323.

## Supplementary Information

## Acknowledgements

We thank the Wysocka lab for providing *Mll3/4* DKO mESCs and their WT controls. We gratefully acknowledge the following research resources at St. Jude Children's Research Hospital for their services: the Hartwell Center for ChIP-seq library preparation and sequencing, the center for Proteomics and Metabolomics for conducting mass spectrometry analysis, and Center for Applied Bioinformatics for bioinformatic analysis. The mouse *α*-Actin monoclonal hybridoma antibody (JLA20) developed by the University of Iowa was obtained from the Developmental Studies Hybridoma Bank, created by the NICHD of the NIH and maintained at The University of Iowa, Department of Biology, Iowa City, IA 52242. We thank W Rosikiewicz and everyone in the Schneider and Herz labs for their helpful comments and critical reading of the manuscript. This work was funded by a transition to independence grant from the National Institutes of Health/National Cancer Institute (R00CA181506 to H-M Herz, P30CA021765 to SM Pruett-Miller); and the American Lebanese Syrian Associated Charities (ALSAC). The content is solely the responsibility of the authors and does not necessarily represent the official views of the National Institutes of Health.

### Author Contributions

X Wang: conceptualization, formal analysis, validation, investigation, visualization, methodology, project administration, and writing—original draft, review, and editing.
W Rosikiewicz: formal analysis, validation, investigation, visualization, methodology, and writing—review and editing.
Y Sedkov: formal analysis, validation, investigation, visualization, and methodology.
B Mondal: formal analysis, validation, investigation, visualization, methodology, and writing—review and editing.
T Martinez: formal analysis, validation, investigation, visualization, and methodology.
S Kallappagoudar: formal analysis, validation, investigation, visualization, methodology, and writing—review and editing.
A Tvardovskiy: formal analysis, validation, investigation, visualization, methodology, and writing—review and editing.
R Bajpai: formal analysis, validation, investigation, visualization, and methodology.
B Xu: formal analysis, validation, investigation, visualization, and methodology.
SM Pruett-Miller: formal analysis, supervision, validation, investigation, visualization, and methodology.
R Schneider: formal analysis, validation, investigation, visualization, methodology, and writing—review and editing.
H-M Herz: conceptualization, formal analysis, supervision, funding acquisition, validation, investigation, visualization, methodology, project administration, and writing—original draft, review, and editing.

### Conflict of Interest Statement

The authors declare that they have no conflict of interest.

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
