## [Reviewer comments · Life Science Alliance]

Life Science Alliance

The MLL3/4 complexes and MiDAC co-regulate H4K20ac to control a specific gene expression program

Xiaokang Wang, Wojciech Rosikiewicz, Yurii Sedkov, Baisakhi Mondal, Tanner Martinez, Satish Kallappagoudar, Andrey Tvardovskiy, Richa Bajpai, Beisi Xu, Shondra Pruett-Miller, Robert Schneider, and Hans-Martin Herz

DOI: 10.26508/lsa.202201572

Corresponding author(s): Hans-Martin Herz, St. Jude Children's Research Hospital and Xiaokang Wang, St. Jude Children's Research Hospital

Review Timeline:

Submission Date:	2022-06-21
Editorial Decision:	2022-06-23
Revision Received:	2022-06-24
Accepted:	2022-06-27

Transaction Report:

Please note that the manuscript was reviewed at Review Commons and these reports were taken into account in the decision-making process at Life Science Alliance.

Review
COMMONS

June 23, 2022

RE: Life Science Alliance Manuscript #LSA-2022-01572

Dr. Hans-Martin Herz
St. Jude Children's Research Hospital
262 Danny Thomas Place
Memphis, TN 38105-3678

Dear Dr. Herz,

Thank you for submitting your revised manuscript entitled "The MLL3/4 complexes and MiDAC act antagonistically as genome-wide regulators of H4K20ac to control a specific gene expression program". We would be happy to publish your paper in Life Science Alliance pending final revisions necessary to meet our formatting guidelines.

- please provide your manuscript text file as an editable doc file
- please upload your main and supplementary figures as single files
- please provide your table files as editable doc or excel files
- please add a Running title, alternate abstract/summary blurb, category, and a Twitter handle of your host institute/organization as well as your own or/and one of the authors in our system
- please add the author contributions to the main manuscript text

Figure Check:

- please include sizes next to blots
- Figure 1D: it looks like a splice was made after column number 27. If so, please indicate with a line and mention in figure legend.

A. FINAL FILES:

B. MANUSCRIPT ORGANIZATION AND FORMATTING:

Sincerely,

Revision Plan

1. General Statements [optional]

This section is optional. Insert here any general statements you wish to make about the goal of the study or about the reviews.

We thank the reviewers for recognizing our contribution to better understand the role of the understudied histone deacetylase complex MiDAC and for acknowledging our work as highly relevant for the understanding of chromatin modification pathways in mammalian cells. The physiological role of MiDAC remains enigmatic and has only very recently been started to be explored by us and others. Work on MiDAC has particularly gained in interest in the last few years as also noted by reviewer 1. Our study for the first time establishes a molecular and mechanistic link between MiDAC and the MLL3/4 complexes. We demonstrate that UTX, a complex-specific subunit of the MLL3/4 complexes, bridges the association of the MLL3/4 complexes and MiDAC by interacting with ELMSAN1, a scaffolding subunit of MiDAC. Our genome-wide and transcriptome-wide analyses indicate that these two complexes functionally relate as antagonistic regulators of H4K20ac to control a specific gene expression program. Considering the prominent role that components of the MLL3/4 complexes such as UTX, MLL3 and MLL4 play in tumor suppression and oncogenesis, this functional link between MiDAC and the MLL3/4 complexes might be leveraged to develop therapeutic approaches to combat cancers that are marked by mutations or misexpression of UTX, MLL3 and MLL4.

Revision Plan

2. Description of the planned revisions

Insert here a point-by-point reply that explains what revisions, additional experimentations and analyses are planned to address the points raised by the referees.

3. Description of the revisions that have already been incorporated in the transferred manuscript

Please insert a point-by-point reply describing the revisions that were already carried out and included in the transferred manuscript. If no revisions have been carried out yet, please leave this section empty.

Reviewer 1:

1. Question: “The conclusion that MiDAC is negative genome-wide regulator of H4K20 acetylation is not convincing as the authors only assessed H4K20ac at Dnttip1 sites. For this conclusion the authors need to show that MiDAC is present at all H4K20ac sites, and that Dnttip1 loss increases H4K20ac at all of its sites.”

Answer: We are grateful to reviewer 1 for his/her insightful suggestion and have now reanalyzed our data accordingly and incorporated the corresponding heatmaps in the revised version of our manuscript. We observe an enrichment of DNNTIP1 at nearly all H4K20ac sites in WT mESCs and this enrichment of DNNTIP1 is lost in *Dnttip1* KO mESCs (Fig S5A). Furthermore, we also find that H4K20ac is increased at the majority of all H4K20ac sites in *Dnttip1* KO compared to WT mESCs (Fig S5B). Based on these analyses, combined with our mass spec results presented in this manuscript (Fig 2) and previously published western blot results for H4K20ac in WT and *Dnttip1* KO mESCs (Mondal et al 2020) we believe the conclusion that MiDAC is a negative genome-wide regulator of H4K20 acetylation is convincing. However, we cannot exclude the possibility that the observed changes in H4K20ac are indirect and might be controlled by other histone deacetylase complexes or histone acetyltransferases as a result of lost MiDAC function. Thus, we have also elaborated on these possibilities in more detail in the discussion section.

2. Question: “Related to this is the presentation of the histone modification mass spec results, which only focus on the H4K20-derived peptide. Can the authors exclude Dnttip1 effects of other histone acetylations?”

Answer: We very much appreciate the reviewer’s thoughtful question. In fact, our mass spec approach also covered other histone acetylation marks on the N-terminal tails of histones H3 and H4 (Fig 2A). H4K20ac was the histone acetylation mark that displayed by far the strongest

Revision Plan

increase in *Dnmtip1* KO compared to WT mESCs (Fig 2A). Except for H3K27ac and H4K16ac which both showed a decrease in acetylation across two independent *Dnmtip1* KO clones, all other acetylation marks besides H4K20ac were not or only mildly affected (Fig 2A). Because H4K20ac was the mark with the strongest acetylation increase and its role in transcription regulation or other processes has not been studied to date we chose to focus on H4K20ac. As requested by the reviewer we have now also included the mass spec results for all these other tested histone acetylation marks and moved the mass spec results that previously were displayed in Figure S4 to incorporate them together with the new mass spec data into a new main figure (Fig 2).

3. Question: “The conclusion of Figure 1E that FLAG-TRErF1 expressing cells could not (purify UTX) is compromised by differences in ELMSAN1 and TRERF1 expression levels. The authors should weaken their conclusion to reflect this.”

Answer: We found that FLAG-TRErF1 is hard to express at high levels. We have thus weakened our conclusion in the revised version of this manuscript as suggested by the reviewer to leave room for the possibility that TRERF1 might also be able to interact with UTX.

Reviewer 2:

1. Question: “Surprisingly, the authors did not show any data on the physical association between endogenous UTX and ELMSAN1/DNMTIP1 in cells. It also remains unclear whether endogenous UTX and MLL3/MLL4 physically associate with MiDAC complex in cells, which raises the possibility that the overexpressed UTX aberrantly associates with MiDAC.”

Answer: We are grateful for the reviewer’s helpful comment. We have also performed UTX and DNMTIP1 IPs against the endogenous proteins in WT and *MLL3/4* DKO mESCs. The results show that the UTX antibody could immunoprecipitate UTX and the MiDAC subunits DNMTIP1, ELMSAN1, and HDAC1 (Fig S2). Furthermore, our DNMTIP1 IP identified DNMTIP1, ELMSAN1, and HDAC1 as members of MiDAC along with UTX (Fig S2). Thus, these results show the physical association between UTX and ELMSAN1/DNMTIP1 under endogenous conditions and exclude the possibility that overexpressed UTX aberrantly associates with MiDAC.

2. Question: “Finally, the authors show that MLL3/MLL4 are required for the genome-wide binding of DNMTIP1 (and presumably MiDAC) and the genome-wide enrichment of H4K20ac, which seems to contradict with the earlier data that DNMTIP1 loss increases H4K20ac.”

“It’s also unclear how the authors reach the conclusion that MiDAC and MLL3/4 complexes oppose each other’s function.”

Revision Plan

Answer: We thank the reviewer for raising these questions. Our conclusion is based on several observations. Most notably the strong opposite effects on histone marks such as H4K20ac and H3K4me2 when the function of MiDAC or the MLL3/4 complexes is abrogated (Figs 3B and C, and 5B and C) and the genome-wide increase in UTX and MLL4 enrichment we observe when MiDAC function is lost (Figs 3C and D, and 4A and B). These findings combined with the opposing transcriptional activities of MiDAC and the MLL3/4 complexes (Fig 6A and B) led us to the conclusion that the MLL3/4 complexes and MiDAC act antagonistically as genome-wide regulators of H4K20ac to control a specific gene expression program. We concur with the reviewer that the reduction in DNTP1 enrichment in *MLL3/4* DKO mESCs is currently difficult to explain. We refer to the discussion section where we have provided multiple potential explanations for this puzzling finding.

3. Question: “Figure 1 panel D data suggests that FLAG-UTX exists in two different complexes: MiDAC and MLL3/MLL4, which does not support the model in Figure 1F.”

Answer: The reviewer's keen observation of two different UTX-containing complexes in Figure 1D can be explained in two different ways. The first possibility is that UTX associates with MiDAC as part of the MLL3/4 complexes (Figure 1D, red box) and also independent of the MLL3/4 complexes (Figure 1D, blue box). Alternatively, the association of UTX with MiDAC outside the MLL3/4 complexes (Figure 1D, blue box) could be an artefact that results from UTX overexpression. The overabundance of UTX due to overexpression might not permit incorporation of a significant portion of UTX into the MLL3/4 complexes because of the limited endogenous amounts of other subunits of the MLL3/4 complexes. Because UTX constitutes the bridging factor between MiDAC and the MLL3/4 complexes this UTX “surplus” (outside the MLL3/4 complexes) is still able to interact with MiDAC. In this scenario the UTX-MiDAC interaction independent of the MLL3/4 complexes (Figure 1D, blue box) would not represent a physiologically relevant complex. We favor the second possibility because UTX is highly unstable in the absence of MLL3/4 (Figs S2; S4B; and S7). This dependency of UTX on MLL3/4 makes it likely that most if not all functions of UTX are carried out through the MLL3/4 complexes and that the observed interaction of overexpressed UTX with MiDAC outside the MLL3/4 complexes does not represent a physiologically relevant complex. We have now further clarified this point in the discussion section.

4. Description of analyses that authors prefer not to carry out

Please include a point-by-point response explaining why some of the requested data or additional analyses might not be necessary or cannot be provided within the scope of a revision. This can be due to time or resource limitations or in case of disagreement about the necessity of such additional data given the scope of the study. Please leave empty if not applicable.

Reviewer 1:

Revision Plan

1. **Question:** “While Mll3/Mll4 loss has the opposite effect on H4K20ac as Dnttip1 loss, the authors do not provide evidence for the responsible histone acetylase for this effect. Can they exclude that the p300/CBP enzymes are responsible, as other colleagues found that the MLL4 complex collaborates with these HATs? Without a better insight into substrate specificity of MiDAC and the responsible H4K20 acetylase, this work remains rather descriptive and lacks a clear mechanistic explanation of the proposed antagonism.”

“This work would benefit from ChIPseq experiments focussing on p300/CBP in the Mll3/Mll4-KO and Dnttip1-KO mESCs. In addition, the interpretation would be helped by ChIPseq for HDAC1/HDAC2.”

“Possibly, biochemical experiments could help to determine the H4K20 specificity of MiDAC. The latter would involve expression purification and HDAC assays, which could easily take 3-6 months, but the proposed ChIPseq experiments can be completed in 1-3 months.”

Answer: We agree with the reviewer that identifying and investigating the role of the histone acetylase(s) responsible for the implementation of H4K20ac would be a worthwhile undertaking. However, we feel that this would be beyond the scope of this study which focuses on the relationship between MiDAC and the MLL3/4 complexes. The histone acetylases p300/CBP have been known to collaborate with the MLL3/4 complexes in enhancer activation both in *Drosophila* and mammals (Lai et al 2017, Tie et al 2012, Wang et al 2016, Wang et al 2017). For example in *MLL3/4* DKO cells, p300/CBP recruitment is reduced at enhancer regions, suggesting that p300/CBP recruitment is dependent on the MLL3/4 complexes (Lai et al., 2017; Wang et al., 2016). Based on these previous publications we anticipate a reduction in p300/CBP enrichment on many enhancers and potentially other genomic elements in *Mll3/4* DKO mESCs. Therefore, if p300/CBP were the responsible H4K20 acetylases their reduced enrichment in *Mll3/4* DKO mESCs could result in decreased H4K20ac (Fig 5B).

However, the ChIP-seq results for p300/CBP in *Mll3/4* DKO and *Dnttip1* KO mESCs are already predictable based on the loss and increase of UTX/MLL4 binding in *Mll3/4* DKO and *Dnttip1* KO mESCs respectively as published by other groups or shown by us in this manuscript. Furthermore, even if we would perform p300/CBP ChIP-seq in *Mll3/4* DKO and *Dnttip1* KO mESCs we still would not be able to draw the conclusion that p300/CBP directly implements H4K20ac as the evidence would only be correlative. Investigating the substrate specificity of MiDAC or p300/CBP towards H4K20ac would require extensive efforts and would have to include structural studies involving NMR or cryo-EM or as suggested by the reviewer biochemical assays which would necessitate purification of MiDAC and p300/CBP and purification and assembly of various acetylated nucleosomal substrates because histone deacetylase complexes and histone acetylases act promiscuously and not specifically on acetylated histone peptides. Experiments of this nature could easily extend over a period of several years, provide enough new information for an independent manuscript and are thus beyond the scope of this study.

Revision Plan

We have also performed HDAC1 ChIP-seq in WT and *Dnmtip1* KO mESCs and, consistent with a loss in MiDAC function, observe a mild reduction in HDAC1 enrichment at many MiDAC-bound sites in *Dnmtip1* KO mESCs (data not shown). This is not surprising considering that HDAC1/2 exist in at least four HDAC1/2-containing complexes (see introduction of this manuscript) and that the other HDAC1/2-containing complexes can still be recruited to chromatin in the absence of MiDAC. However, we do not think that addition of this data would aid with any of the conclusions drawn in this manuscript.

Reviewer 2:

1. Question: “The knockout of DNTTIP1 also led to increased binding of UTX/MLL4 on a subset of regions, although it is unclear whether this is secondary to the increase of H4K20ac. The authors need to do ATAC-seq examine chromatin opening after DNTTIP1 knockout.”

Answer: We agree with the reviewer that ATAC-seq is an outstanding way to assess chromatin opening and that increased chromatin opening due to higher enrichment of H4K20ac might contribute to or cause the increase in UTX/MLL4 binding we observe in *Dnmtip1* KO mESCs (Figs 3C and D, and 4A-C). However other possibilities also exist. As mentioned in the discussion section previous studies have shown that the PHD6 domain of MLL4 is able to recognize and bind H4K16ac (Zhang et al 2019). Therefore, it is possible that other PHD domains or regions within MLL3/4 might recognize or bind H4K20ac to enhance recruitment of MLL3/4. Furthermore, performing ATAC-seq would only provide correlative evidence between chromatin opening and UTX/MLL4 binding in *Dnmtip1* KO mESCs but would ultimately not be able to answer the question whether chromatin opening was causing increased UTX/MLL4 enrichment in *Dnmtip1* KO mESCs or not.

References

Lai B, Lee JE, Jang Y, Wang L, Peng W, Ge K. 2017. Mll3/ml14 are required for cbp/p300 binding on enhancers and super-enhancer formation in brown adipogenesis. *Nucleic acids research*. 45(11):6388-6403. doi:10.1093/nar/gkx234

Mondal B, Jin H, Kallappagoudar S, Sedkov Y, Martinez T, Sentmanat MF, Poet GJ, Li C, Fan Y, Pruett-Miller SM, et al. 2020. The histone deacetylase complex midac regulates a neurodevelopmental gene expression program to control neurite outgrowth. *eLife*. 9 doi:10.7554/eLife.57519

Tie F, Banerjee R, Conrad PA, Scacheri PC, Harte PJ. 2012. The histone demethylase utx and the chromatin remodeler brm bind directly to drosophila cbp and modulate its acetylation of histone h3 lysine 27. *Molecular and cellular biology*. doi:10.1128/MCB.06392-11

Revision Plan

Wang C, Lee JE, Lai B, Macfarlan TS, Xu S, Zhuang L, Liu C, Peng W, Ge K. 2016. Enhancer priming by h3k4 methyltransferase mll4 controls cell fate transition. *Proceedings of the National Academy of Sciences of the United States of America*. 113(42):11871-11876. doi:10.1073/pnas.1606857113

Wang SP, Tang Z, Chen CW, Shimada M, Koche RP, Wang LH, Nakadai T, Chramiec A, Krivtsov AV, Armstrong SA, et al. 2017. A utx-mll4-p300 transcriptional regulatory network coordinately shapes active enhancer landscapes for eliciting transcription. *Molecular cell*. 67(2):308-321 e306. doi:10.1016/j.molcel.2017.06.028

Zhang Y, Jang Y, Lee JE, Ahn J, Xu L, Holden MR, Cornett EM, Krajewski K, Klein BJ, Wang SP, et al. 2019. Selective binding of the phd6 finger of mll4 to histone h4k16ac links mll4 and mof. *Nat Commun*. 10(1):2314. doi:10.1038/s41467-019-10324-8

June 27, 2022

RE: Life Science Alliance Manuscript #LSA-2022-01572R

Dr. Hans-Martin Herz
St. Jude Children's Research Hospital
262 Danny Thomas Place
Memphis, TN 38105-3678

Dear Dr. Herz,

Thank you for submitting your Research Article entitled "The MLL3/4 complexes and MiDAC co-regulate H4K20ac to control a specific gene expression program". It is a pleasure to let you know that your manuscript is now accepted for publication in Life Science Alliance. Congratulations on this interesting work.

DISTRIBUTION OF MATERIALS:

Again, congratulations on a very nice paper. I hope you found the review process to be constructive and are pleased with how the manuscript was handled editorially. We look forward to future exciting submissions from your lab.

Sincerely,
